# Higher- and lower-order personality traits and cluster subtypes in social anxiety disorder

Mădălina Elena Costache[1], Andreas Frick[2], Kristoffer Månsson[1,3,4,5], Jonas Engman[1], Vanda Faria[1,6,7], Olof Hjorth[1], Johanna M. Hoppe[1], Malin Gingnell[1,8], Örjan Frans[1], Johannes Björkstrand[1,9], Jörgen Rosén[1], Iman Alaie[1,10], Fredrik Åhs[11], Clas Linnman[12], Kurt Wahlstedt[1], Maria Tillfors[13], Ina Marteinsdottir[14], Mats Fredrikson[15], Tomas Furmark[1]*

1 Department of Psychology, Uppsala University, Uppsala, Sweden, 2 The Beijer Laboratory, Department of Neuroscience, Uppsala University, Uppsala, Sweden, 3 Centre for Psychiatry Research, Department of Clinical Neuroscience, Karolinska Institutet, & Stockholm Health Care Services, Stockholm County Council, Stockholm, Sweden, 4 Max Planck UCL Centre for Computational Psychiatry and Ageing Research, Berlin, Germany and London, United Kingdom, 5 Center for Lifespan Psychology, Max Planck Institute for Human Development, Berlin, Germany, 6 Center for Pain and The Brain, Department of Anesthesiology, Harvard Medical School, Boston Children's Hospital, Perioperative and Pain Medicine, Boston, MA, United States of America, 7 Department of Otorhinolaryngology, Smell & Taste Clinic, TU Dresden, Dresden, Germany, 8 Department of Neuroscience, Uppsala University, Uppsala, Sweden, 9 Department of Psychology, Lund University, Lund, Sweden, 10 Department of Neuroscience, Child and Adolescent Psychiatry, Uppsala University, Uppsala, Sweden, 11 Department of Psychology and Social Work, Mid Sweden University, Östersund, Sweden, 12 Harvard Medical School, Spaulding Rehabilitation Hospital, Boston, MA, United States of America, 13 Department of Social and Psychological Studies, Karlstad University, Karlstad, Sweden, 14 Department of Clinical and Experimental Medicine, Linköping University, Linköping, Sweden, 15 Department of Clinical Neuroscience, Karolinska Institutet, Stockholm, Sweden

* tomas.furmark@psyk.uu.se

**Data Availability Statement:** The data underlying the results presented in the study are available from https://www.psyk.uu.se/forskning/

## Abstract

Social anxiety disorder (SAD) can come in different forms, presenting problems for diagnostic classification. Here, we examined personality traits in a large sample of patients (N = 265) diagnosed with SAD in comparison to healthy controls (N = 164) by use of the Revised NEO Personality Inventory (NEO-PI-R) and Karolinska Scales of Personality (KSP). In addition, we identified subtypes of SAD based on cluster analysis of the NEO-PI-R Big Five personality dimensions. Significant group differences in personality traits between patients and controls were noted on all Big Five dimensions except agreeableness. Group differences were further noted on most lower-order facets of NEO-PI-R, and nearly all KSP variables. A logistic regression analysis showed, however, that only neuroticism and extraversion remained significant independent predictors of patient/control group when controlling for the effects of the other Big Five dimensions. Also, only neuroticism and extraversion yielded large effect sizes when SAD patients were compared to Swedish normative data for the NEO-PI-R. A two-step cluster analysis resulted in three separate clusters labelled *Prototypical* (33%), *Introvert-Conscientious* (29%), and *Instable-Open* (38%) SAD. Individuals in the *Prototypical* cluster deviated most on the Big Five dimensions and they were at the most severe end in profile analyses of social anxiety, self-rated fear during public speaking, trait anxiety, and anxiety-related KSP variables. While additional studies are needed to determine if personality subtypes in SAD differ in etiological and treatment-related factors, the

forskargrupper/uppsala-affective-neuroscience-group/

**Funding:** Supported by the Swedish Research Council (grant 2016-0228) and Riksbankens Jubileumsfond - the Swedish Foundation for Research in Social Sciences and the Humanities (grant P17-0639:1) https://www.vr.se/ https://www.rj.se/ The funders had no role in study design, data collection and analysis, decision to publish, or preparation of the manuscript.

**Competing interests:** The authors have declared that no competing interests exist.

present results demonstrate considerable personality heterogeneity in socially anxious individuals, further underscoring that SAD is a multidimensional disorder.

# Introduction

Social anxiety disorder (SAD) is one of the most common psychiatric disorders [1] characterized by a persistent and over-whelming fear of being negatively evaluated in one or more social or interactional situation [2]. It is associated with considerable individual suffering [3], large societal costs [4,5] and typically follows a chronic course if left untreated [6]. Cognitive behavioral therapy (CBT), serotonin reuptake inhibitors (SSRIs) and serotonin-noradrenaline reuptake inhibitors (SNRIs) are first-line treatment options for SAD [7,8]. Although these treatments are helpful, as many as 40–50% of patients have been reported to be either treatment resistant or not responding sufficiently [9]. Several factors, like variations in symptom profile and comorbidity of personality disorders, may underlie this and more research is needed to better understand the etiology and relevant treatment approaches of SAD. Social anxiety can be studied, not only as a disorder, but also as one or more dispositional traits involving emotional discomfort and social withdrawal [10]. Spence and Rapee suggested that social anxiety may be a personality-like construct while SAD diagnosis reflects an interaction between social anxiety and the degree of impairment such anxiety imposes in life [11]. Maladaptive personality traits may have a large impact on psychosocial functioning and, hence, the course and expression of psychiatric disorders. Moreover, disorders and traits may share a common etiology [12] and personality traits could be predictive of treatment outcome [13,14]. Deciphering the complex relationships between basic personality traits and SAD is therefore theoretically and clinically important.

The revised NEO Personality Inventory (NEO-PI-R) provides comprehensive assessment of personality dimensions, and their underlying facets, based on the five-factor model of personality i.e., the "Big Five" neuroticism, extraversion, openness, agreeableness, and conscientiousness [15]. Previous studies have reported that SAD is associated high scores of neuroticism and low scores of extraversion [16–19]. Marteinsdottir and colleagues [20] assessed personality traits in a sample of Swedish untreated SAD individuals by use of another common personality inventory, the Karolinska Scales of Personality; KSP [21]. In comparison to normative data, the SAD sample scored higher on the KSP scales related to vulnerability for anxiety, detachment, irritability, and indirect aggression, and lower on socialization and social desirability. SAD patients with comorbid avoidant personality disorder scored higher on inhibition of aggression and psychic anxiety [20]. Personality dimensions in SAD have also been evaluated by means of the Temperament and Character Inventory (TCI) [22]. Clinical SAD samples have then exhibited significantly higher harm-avoidance, and significantly lower self-directedness, persistence, cooperativeness, self-transcendence, and novelty seeking when compared to healthy participants [23,24]. Notably, sample sizes in these studies have been limited, generally not exceeding N = 60. More studies with larger samples are needed to clarify the crucial personality components associated with SAD, including higher-order dimensions as well as lower-order facets. Also, little is known regarding the impact of such personality components on subtypes of SAD.

The heterogeneity of SAD has been widely acknowledged [25] and several subtypes have been proposed over the years. However, empirical research into SAD subtypes has yielded mixed findings and a resultant general lack of consensus, partly reflecting use of different

statistical methods and samples [26]. Social anxiety may extend to a broad range of situations and the generalized subtype of SAD was introduced in DSM-III-R as a descriptor of individuals who fear most social situations. The residual category has often been referred to as "nongeneralized". However, anxiety reactions may also be limited to one or two social situations, typically performance situations like public speaking. Heimberg and colleagues [27] proposed that "circumscribed" SAD should be added to the generalized and nongeneralized subtypes, and other labels have also been suggested such as "specific", "discrete", and "limited interactional" SAD [27,28]. Blöte and colleagues argued that public speaking anxiety is a distinct subtype, different from other subtypes [29]. In the current version of DSM, i.e. DSM-5, generalized SAD has been replaced by "performance type" as the only subtype specifier, although this may not do justice to the complexity of the issue.

As in psychiatry in general, it has been debated whether SAD subtypes are best described as categories or dimensions. Support for a dimensional mild-moderate-severe subtype distribution was found in a cluster analytic study of SAD in a community sample [28] and other empirical studies have also concluded that the heterogeneity of SAD should be seen as a continuum of severity, greater number of social fears being associated with greater disability [30–33]. On the other hand, subgrouping can also be based on the type of social anxiety. The presence of observational vs. interactional anxiety could be a putative qualitative demarcation of SAD subtypes [34]. Using factor analysis in a clinical SAD sample, Perugi and colleagues found support for the existence of five types of social anxiety: interpersonal anxiety, formal speaking anxiety, stranger-authority anxiety, eating and drinking while being observed, and anxiety of doing something while observed [35]. Moreover, studies have found evidence of qualitatively different SAD subgroups based on Cloninger's temperamental characteristics [22]. By use of cluster or latent class analysis, researchers have identified not only a prototypical SAD subgroup characterized by high harm-avoidance and low novelty seeking, but also an anxious-impulsive subtype scoring high on novelty seeking [36–39]. While individuals in the former group show behavioral inhibition and risk aversion, individuals in the latter exhibit an atypical pattern of risk-prone approach behaviors while still being highly anxious. From a theoretical perspective, Hofmann and colleagues have suggested that subtypes of SAD vary across six dimensions: fearfulness, anxiousness, shyness, self-consciousness, submissiveness, and anger [25]. Notably, these dimensions overlap considerably with neuroticism and extraversion facets that can be assessed with instruments like the NEO-PI-R.

The controversies around SAD subtyping bear strong resemblance with debates in personality research concerning the usefulness of qualitative types vs. quantitative traits and person-centered vs. variable-centered approaches [40,41]. There have been attempts to quantify personality types from trait instruments like the NEO-PI-R [42], and according to a widely-cited typology, people may fall into three distinct categories: 'resilient', 'overcontrolled' or 'undercontrolled', e.g. [40]. Resilients have below average scores on neuroticism and above average or intermediate scores on the remaining four dimensions; overcontrollers score high in neuroticism and low in extraversion whereas undercontrollers have low scores in conscientiousness and agreeableness [43]. Recently Gerlach et al. [44] found evidence of four robust personality types in a Big Five data set comprising 1.5 million individuals. These were labelled "average", "self-centred", "reserved" and "role model" respectively, the latter showing resemblance with "resilient" [44]. It is not well understood how SAD subgroups compare with these personality types. Presumably, prototypical SAD individuals are overcontrollers but this may not be true for the anxious-impulsive SAD subtype [36–39]. Anyhow, studies exploring subtypes of SAD by personality inventories are scant and, to our knowledge, no previous study has evaluated potential subtypes of SAD derived from the widely researched Big Five personality dimensions.

As social anxiety may be conceptually intertwined with several personality components, the principal aim of the present study was to examine personality traits in a large sample of individuals diagnosed with SAD (N = 265), in comparison to healthy controls (N = 164) and Swedish normative data, by use of the NEO-PI-R and KSP instruments. We expected elevated neuroticism and lower extraversion on the NEO-PI-R, as well as higher scores on KSP items related to anxiety and behavioral inhibition, in SAD individuals. Further aims were to explore subtypes of SAD by use of cluster analysis of the Big Five personality dimensions, and to compare the personality types with respect to other clinical variables including social anxiety symptom severity, interaction anxiety, trait anxiety, KSP scales and affective ratings during a public speaking challenge.

## Methods

### Participants characteristics and general study set-up

In total, 265 patients [117 men, 148 women; mean age (SD): 33.5 (10.3) years] diagnosed with DSM-IV SAD [45] and 164 healthy controls [82 men, 82 women; mean age: 30.9 (9.9) years], answered paper-and-pen version of the personality scales NEO-PI-R and KSP. All participants were volunteers in neuroimaging treatment trials, data being collected from 1998 to 2018, as described elsewhere [46–54]. NEO-PI-R data were collected from trials conducted from 2003 and onwards. All studies were approved by the Regional Ethical Review Board in Uppsala and all participants provided written informed consent. The personality forms were filled out in the home-environment before neuroimaging assessment and any subsequent treatment.

Patients with SAD were recruited mainly through media advertisements while healthy controls answered both to public billboards at Uppsala University and newspaper advertisements. The psychiatric status was assessed either by a clinical psychologist or a psychiatrist, who administered the anxiety disorders section of Structured Clinical Interview for DSM-IV (SCID-I) [55] and the Mini International Neuropsychiatric Interview [56]. The complete SCID-I and SCID-II interviews were administered in one study [54]. Participants underwent a medical check-up and were considered physically healthy. All patients met the criteria for a primary SAD diagnosis according to DSM-IV [45] with marked fear of social situations including public speaking. Forty-four (17%) presented one comorbid secondary Axis I disorder, 21 (8%) presented two comorbidities and 2 patients (0.8%) had three comorbidities. Comorbid conditions included generalized anxiety disorder, specific phobia, obsessive-compulsive disorder, panic disorder with or without agoraphobia, post-traumatic stress disorder and mild major depressive disorder. None of the controls fulfilled the screening criteria for SAD or any other psychiatric condition.

Exclusion criteria were: previous or current neurological and somatic illnesses, current predominant axis I mental disorder other than SAD (e.g. bipolar or severe major depressive disorder, psychosis), pregnancy, menopause, psychological or psychotropic treatment that was ongoing or had ended within the previous three months, alcohol and narcotics addiction or abuse, age outside the range of 18–65, or other characteristics that could be expected to interfere with the original neuroimaging study such as claustrophobia or metal implants [46–54].

### Personality instruments

Personality traits were measured by Swedish versions of the NEO-PI-R [15] and KSP [21]. The NEO-PI-R consists of 240 Likert-scale items, rated from 0 ("absolutely disagree) to 4 ("absolutely agree). It is a widely recognized instrument developed to improve the general comprehension of personality in adults by assessing five factors (neuroticism, extraversion, agreeableness, openness to experience, and conscientiousness), and six categories (facets) of

each one of the five higher-order traits. Cronbach's alpha values for NEO-PI-R factors in the present study were: neuroticism 0.92, extraversion 0.86, openness 0.75, conscientiousness 0.80, and agreeableness 0.62.

The KSP inventory was created with the aim of quantifying imperative dimensions of personality or temperament, based on psychobiological theories and research [57–59]. The instrument is composed of 135 items grouped into 15 scales: five scales assess propensity to experience anxiety states (somatic anxiety, psychic anxiety, muscular tension, psychasthenia, and inhibition of aggression), three dimensions are related to susceptibility for behavioral disinhibition (impulsivity, monotony avoidance, and detachment), and the remaining scales are mainly associated to hostility and aggression (indirect and verbal aggression, irritability, suspicion, guilt, socialization, and social desirability). In the present study, internal consistency ranged from 0.61 for hostility to 0.92 for anxiety dimensions.

## Other instruments

Additional clinical measures were used to compare clusters of SAD individuals. Social anxiety symptom severity was measured primarily by the Liebowitz Social Anxiety Scale, LSAS [60,61]. Social interaction anxiety was measured by the Social Interaction Anxiety Scale, SIAS [62]. Trait anxiety was assessed by Spielberger's State-trait Anxiety Inventory, STAI-T [63]. Moreover, self-rated fear and distress were assessed with 0–100 (min-max) scales during a public speaking behavioral test administered in conjunction with the neuroimaging trial, see e.g., [49,50,52,54]. Because the public speaking challenge was administered within the scanner for PET trials, but outside the scanner for fMRI trials, we used type of test as a covariate in group comparisons. Finally, clinician-rated data on severity category (mild/moderate/severe) were retrieved from the diagnostic interview (SCID) forms or, in case of missing information, a severity rating was derived from the Clinical Global Impression–Severity (CGI-S) scale [64], with scores of $\geq 5$ indicating severe, 4 = moderate, and 3 = mild. Diagnostic interview data on DSM-IV subgroup (generalized/nongeneralized SAD), and avoidant personality disorder (yes/no) as assessed with the SCID-II [65] was obtained in a subset (n = 72) of the SAD sample.

## Statistical analyses

Statistical analyses were performed using SPSS Version 25 (IBM SPSS Statistics for Windows, Version 25.0. Armonk, NY: IBM Corp). Independent sample t-tests were run to compare the mean scores between the two groups on both personality scales. Bonferroni adjustment for multiple comparisons was used for Big Five dimensions whereas Holm adjusted alpha levels were applied for NEO-PI-R facets and KSP variables due to the larger number of comparisons. To determine the magnitude of observed significant effects, a between-group effect size was calculated using Cohen's *d* formula [66]. For informatory purposes effect sizes (*d*) were also calculated for SAD vs. normative group comparisons, using Swedish norm data for NEO-PI-R [67] and KSP [68]. Logistic regression analysis including the Big Five personality variables was performed (with a $p<.01$ Bonferroni criterion) to identify independent predictors of group (patient or control).

Two-step cluster analysis with log-likelihood distance measures was used in SPSS for exploratory detection of potentially similar groups of persons with relatively homogenous personality traits [69]. The 15 KSP variables were previously found to represent "lower-order traits" for neuroticism, extraversion, agreeableness, while no representation was found for openness or conscientiousness [68]. Because of this, the NEO-PI-R Big Five dimensions were selected as cluster variables, and the KSP scales as profile variables, in the analysis. One-way analyses of variance (ANOVAs) were performed to ascertain significant differentiation

between the resultant clusters, using a standard level of significance ($p<0.05$) followed by Bonferroni post hoc comparisons, controlling for multiple comparisons.

## Results

### Group differences in demographic characteristics

There were no differences between the SAD patients and healthy controls with respect to gender distribution ($\chi^2 = 1.394; p = .273$). There was a group difference in age ($t = 2.601; df = 427; p = .010$), but age did not correlate with the NEO-PI-R or KSP personality variables, except for weak correlations with *Neuroticism* ($r = −.113, p<.05$), *Openness* ($r = −.138, p<.01$), *Social Desirability* ($r = .190, p<.01$), *Monotony Avoidance* ($r = −.137, p<.05$), and *Detachment* ($r = .193, p<.01$). Controlling for age in the subsequent statistical analyses did not alter any significant result.

### Group differences in the revised NEO personality inventory

In total, 211 SAD patients (91 men, 120 women; mean age ± SD: 32.7 ±10.6 years) and 138 healthy control participants (73 men, 65 women; 30.8 ± 9.9 years) completed the NEO-PI-R self-report. Independent samples t-tests revealed that subjects with SAD had significantly higher scores on neuroticism and significantly lower scores on extraversion, openness, and conscientiousness, with large effect sizes, as compared to healthy controls ($p<.001$)—see Table 1. On facets, there were statistically robust group differences on all lower-order traits of extraversion and neuroticism (S1 Table). For openness and conscientiousness facets, between-group effect sizes varied from moderate to large and significant differences, exceeding the Bonferroni criterion, were found on openness to actions-O4, ideas-O5, and values-O6; competence-C1, dutifulness-C3, and self-discipline-C5. Despite no group difference on the full agreeableness dimension, significant differences were found at the facet level but in mixed directions, with lower trust-A1 and altruism-A3, but higher straightforwardness-A2 and modesty-A5, in patients–see S1 Table.

When comparing SAD patients to Swedish normative data [68] large effect sizes were only noted for neuroticism and extraversion and a moderate effect size for conscientiousness (Table 1). Effect sizes were also large for 8 of the 12 neuroticism and extraversion facets, as well as for self-discipline-C5 (S1 Table). On openness to ideas-O5 and values-O6, patients scored lower than the control sample but higher than the Swedish normative group, whereas patients were steadily lower on openness for actions-O4.

To further evaluate personality dimensions that were independent predictors of group (SAD or control), a logistic regression analysis was conducted. Results showed that only neuroticism and extraversion were robust significant predictors ($p\leq.001$) when all dimensions were included in the statistical model (Table 2). The model explained 83% of the variance, according to Nagelkerke R Square and correctly classified 93% of cases. Hosmer and Lemeshow test indicated adequate goodness of fit ($\chi^2 = 5.536; p = .699$). Variance inflation factors (VIF) were <2.22 indicating no serious multicollinearity. Controlling for age in the model did not alter results, neuroticism and extraversion remaining highly significant ($p < .001$) predictors.

### Group differences in the Karolinska Scales of Personality

The KSP was completed by 217 patients (99 men, 118 women; mean age ± SD 34.1 ±10.6 years) and 123 healthy control subjects (64 men, 59 women; 30.4 ±10.0 years). Significantly higher scores for the SAD sample, in comparison to controls, were noted on psychic anxiety,

**Table 1. Comparisons of social anxiety disorder (SAD) patients and healthy controls (HC) on NEO-PI-R Big Five dimensions.**

|  | SAD N = 211 | HC N = 138 | t | p | d vs. HC | d vs. norms[1] |
|---|---|---|---|---|---|---|
|  | M (SD) | M (SD) |  |  |  |  |
| Neuroticism | 114.23 (23.59) | 60.04 (22.55) | 21.35 | < .001 | 2.35 | 1.57 |
| Extraversion | 80.50 (22.06) | 123.61 (18.07) | -19.14 | < .001 | -2.14 | -1.27 |
| Openness | 107.39 (22.29) | 121.27 (22.42) | -5.67 | < .001 | -0.62 | 0.10 |
| Agreeableness | 131.31 (18.32) | 131.79 (18.24) | -.24 | .812 | -0.03 | 0.06 |
| Conscientiousness | 109.65 (20.96) | 126.33 (20.66) | -7.31 | < .001 | -0.80 | -0.59 |

Bonferroni adjusted $\alpha$ = 0.01; NEO-PI-R = Revised NEO Personality Inventory $d$ = between-group effect size according to Cohen's $d$

[1]SAD in comparison to Swedish norm data [67], (M±SD): N (78.0±22.5), E (107.6±20.7), O (105.2±21.3), A (130.3±17.2), C (121.4±18.8).

somatic anxiety, psychasthenia, inhibition of aggression, detachment, muscular tension, irritability, suspicion, and guilt. Significantly lower scores were noted for socialization, monotony avoidance, impulsivity, social desirability and verbal aggression ($p \leq .005$)–see Table 3. Effect sizes were generally large or very large. Only on indirect aggression, the group difference was non-significant (p = 0.062). Comparing SAD with normative data also confirmed a largely deviant KSP profile in the patient sample although with more conservative estimates of effect size (Table 3). Because of the large number of scales and multicollinearity issues, logistic regression was not used for the KSP. Correlations between KSP scales and NEO-PI-R dimensions are given in S2 Table (SAD sample).

## Two-step cluster analysis of personality types in social anxiety disorder

The 211 SAD patients with complete NEO-PI-R data were included in a two-step cluster analysis using log-likelihood distance measures, Schwarz's Bayesian Criterion (BIC) as validation measure [70], and the Big Five dimensions as cluster variables. This resulted in a three-cluster solution–see Fig 1. The five input variables yielded a silhouette coefficient of 0.3, indicative of fair cluster homogeneity. The variable exhibiting the highest predictor importance, in the creation of the three clusters, was extraversion, followed by neuroticism, conscientiousness and openness (Fig 1A). Based on the subsequent descriptive and profile analyses (see further below), cluster 1 was labelled *Prototypical* (n = 69, 32.7%); cluster 2 *Introvert-Conscientious* (n = 62; 29.4%); and cluster 3 *Instable-Open* (n = 80, 37.9%)–see Fig 1B.

As indicated by separate ANOVA's, significant differences (p < .001) between the three clusters were confirmed for neuroticism (F(2,210) = 51.92; $\eta^2$ = .341), extraversion (F(2,210) = 107.87, η2 = .707), openness (F(2,210) = 60.77; $\eta^2$ = .530), and conscientiousness (F(2,210) = 48.50, $\eta^2$ = .370). All differences remained significant also with healthy controls included in the analyses (Table 4). Differences between clusters at the facet level are listed in S3 Table.

**Table 2. Logistic regression analysis of Revised NEO Personality Inventory personality predictors of diagnostic group, i.e. social anxiety disorder or healthy control.**

|  | β | SE | Wald | p | OR | 95% CI |
|---|---|---|---|---|---|---|
| Neuroticism | .071 | .011 | 42.066 | < .001 | 1.074 | 1.051–1.097 |
| Extraversion | -.076 | .014 | 31.002 | < .001 | .927 | .902 - .952 |
| Openness | -.007 | .012 | .335 | .563 | .993 | .970–1.017 |
| Agreeableness | .029 | .013 | 5.298 | .021 | 1.029 | 1.004–1.055 |
| Conscientiousness | .001 | .012 | .004 | .952 | 1.001 | .978–1.024 |

β = standardized coefficient; CI = confidence interval; SE = standard error; OR = odds ratio; Bonferroni adjusted $\alpha$ = 0.01

**Table 3. Comparison of social anxiety disorder (SAD) patients and Healthy Controls (HC) on the Karolinska Scales of Personality.**

| | SAD N = 217 | HC N = 123 | t | p | d vs. HC | d vs. norms[1] |
|---|---|---|---|---|---|---|
| | M (SD) | M (SD) | | | | |
| Psychic Anxiety | 29.44 (4.97) | 16.81 (4.94) | 22.56 | <0.001 | 2.55 | 1.64 |
| Somatic Anxiety | 23.58 (5.20) | 14.08 (3.82) | 19.26 | <0.001 | 2.08 | 1.32 |
| Psychastenia | 26.06 (4.57) | 18.49 (4.34) | 14.93 | <0.001 | 1.70 | 1.21 |
| Inhibition of Aggression | 29.14 (5.28) | 21.81 (4.02) | 14.36 | <0.001 | 1.56 | 1.17 |
| Detachment | 25.18 (5.15) | 18.22 (3.98) | 13.88 | <0.001 | 1.51 | 0.82 |
| Muscular Tension | 21.52 (5.63) | 14.24 (4.67) | 12.80 | <0.001 | 1.41 | 1.07 |
| Irritability | 12.62 (2.37) | 9.66 (2.28) | 11.24 | <0.001 | 1.27 | 0.48 |
| Suspicion | 11.18 (2.70) | 7.99 (2.31) | 11.48 | <0.001 | 1.27 | 0.66 |
| Socialization | 59.07 (9.11) | 68.91 (9.01) | -9.61 | <0.001 | -1.09 | -0.91 |
| Guilt | 12.51 (2.27) | 10.70 (2.03) | 7.34 | <0.001 | 0.84 | 0.37 |
| Monotony Avoidance | 21.81 (5.32) | 25.76 (4.89) | -6.78 | <0.001 | -0.77 | -0.23 |
| Impulsivity | 20.69 (4.47) | 23.59 (4.48) | -5.73 | <0.001 | -0.65 | 0.52 |
| Social Desirability | 26.72 (3.81) | 28.59 (3.70) | -4.39 | <0.001 | -0.50 | NA |
| Verbal Aggression | 10.61 (2.96) | 11.53 (2.74) | -2.81 | 0.005 | -0.32 | -0.71 |
| Indirect Aggression | 12.12 (2.91) | 11.51 (2.77) | 1.87 | 0.062 | 0.21 | 0.14 |

Holm adjusted $\alpha$ = .025–.0033; d = between-group effect size according to Cohen's d

[1]SAD in comparison to Swedish norm data [68].

Bonferroni post hoc comparisons revealed that all clusters differed significantly from the healthy controls on neuroticism and extraversion, cluster 1 having the most deviant profile–see Fig 2. This cluster was labelled *Prototypical*, to conform with terminology used in other studies [e.g., 36–39]. Although cluster 1 and 2 had comparable levels of low extraversion (significant differences were noticed only on assertiveness-E3), cluster 2 had much lower scores of neuroticism. Additionally, cluster 2 was characterized by significantly higher conscientiousness, with values comparable to the non-clinical group (Table 4, Fig 1C), supporting labelling of this cluster as *Introvert-Conscientious*. With regard to openness, cluster 3 was similar to healthy controls, higher than norms and significantly more open than the other SAD clusters. This cluster also exhibited considerably higher levels of extraversion in comparison to the other SAD clusters, although still lower than in healthy controls (Table 4 and Fig 2). On neuroticism, also referred to as emotional stability, these individuals had significantly higher values than cluster 2 (and controls). Hence, this cluster was labelled *Instable-Open*.

As may be expected, given that no SAD case-control group difference on agreeableness was found, all clusters had similar values as healthy controls on this dimension. However, a somewhat mixed pattern of differences was noted at the facet level (S3 Table). For example, the *Prototypical* cluster showed significantly lower values of trust-A1 but higher values of compliance-A4 and modesty-A5 in comparison to controls. In general, the three clusters differed markedly relative to Swedish normative data, as reflected in effect size estimates, with agreeableness being the only clear exception (Table 4).

## Cluster profile analyses

No difference was found in gender distribution across clusters ($\chi^2$ = 3.79, p = .150). Comparative statistics on six other cluster profile variables are given in Table 5. The ANOVAs indicated differences in mean age, *Introvert-Conscientious* individuals being relatively older (F(2,210) = 4.70, p = .010). The three clusters were significantly differentiated on social anxiety symptom

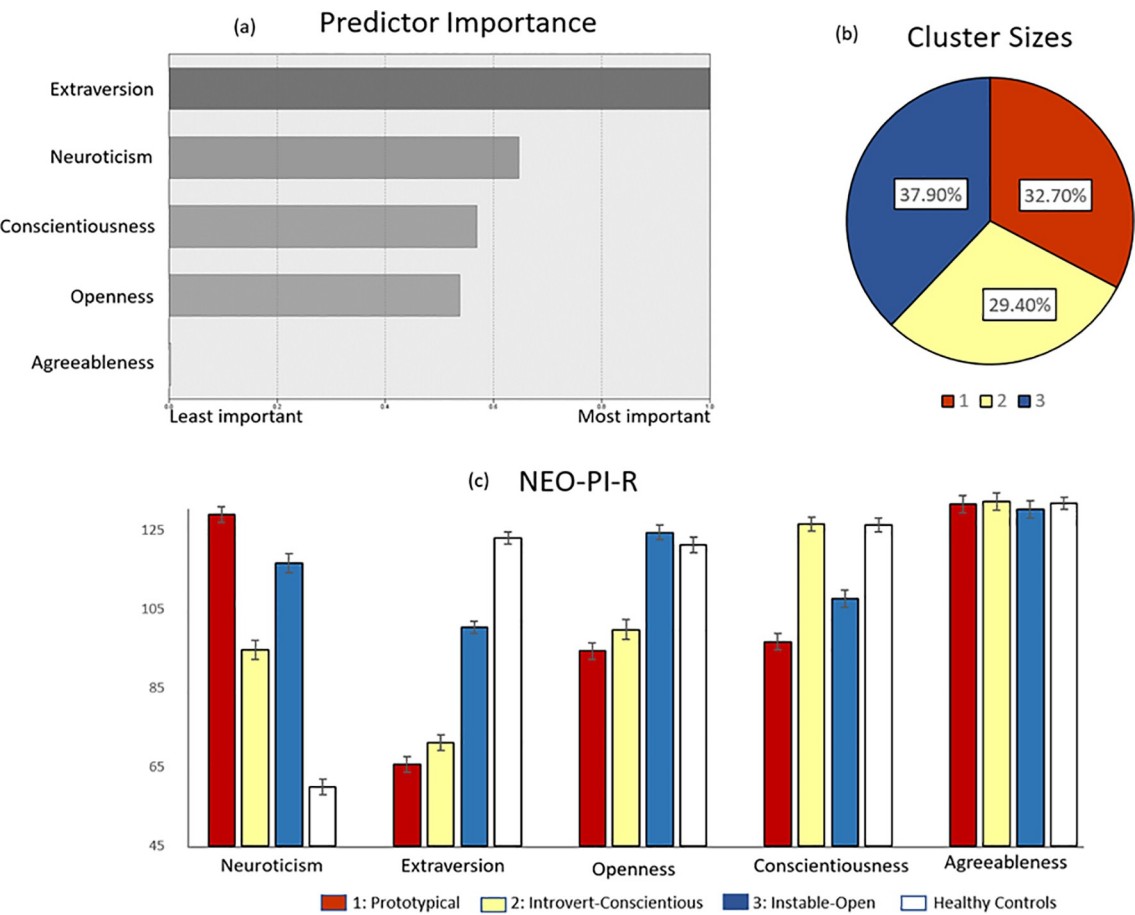

**Fig 1. Cluster analysis solution. (a)** Predictor importance of the five personality dimensions in the two-step cluster analysis with extraversion showing highest importance; **(b)** Distribution of social anxiety disorder (SAD) patients across the three resultant clusters; **(c)** Differences between the three SAD clusters on the five personality dimensions used as cluster variables. Healthy controls (n = 138) are also displayed for informatory purposes. Error bars represent standard errors.

severity (LSAS), social interaction anxiety (SIAS), and trait anxiety (STAI-T)–see Table 5. Post hoc comparisons showed higher trait-anxiety in the *Prototypical* cluster and significantly lower levels of social anxiety (LSAS) and interaction anxiety (SIAS) in the *Instable-Open* cluster relative to the others. ANCOVA, having type of public speaking test (inside/outside scanner) as covariate, also revealed significant differences between clusters in self-rated fear during the test, the *Prototypical* cluster again being at the most severe end.

S4 Table further shows the personality KSP scores across the three clusters and the healthy control group. Cluster 1 *Prototypical* reported the highest levels of psychic anxiety, muscular tension, psychasthenia and inhibition of aggression, in comparison to the other two SAD clusters. The *Introvert-Conscientious* cluster had a less affected profile in terms of social desirability, socialization, and guilt, whereas the *Instable-Open* cluster showed increased levels of monotony avoidance and impulsivity, and decreased detachment indicative of higher extraversion.

Clusters were further compared on clinician-rated data retrieved from the diagnostic interview forms. Clusters did not differ significantly with regard to presence of ($\chi^2$ = 2.20, $df$ = 2, $p$ = .33) or number of (F = .33, df = 2,208, p = .72) current comorbid Axis I conditions. Significant differences across clusters were, however, noted on severity rating i.e., mild/moderate/severe

**Table 4. Mean values (SD) and ANOVA results on the NEO-PI-R Big Five dimensions in three clusters of social anxiety disorder (SAD) patients compared with healthy controls (HC).**

|  | (n = 138) HC | 1 (n = 69) Prototypical | 2 (n = 62) Introvert- Conscientious | 3 (n = 80) Instable-Open | F (3, 348) | P | Post-hoc |
|---|---|---|---|---|---|---|---|
| Neuroticism | 60.04 (22.55) | 128.93 (15.85) | 94.74 (18.54) | 116.66 (22.48) | 221.21 | < .001 | HC<2<3<1 |
| d vs. norms[1] |  | 2.62 | 0.81 | 1.72 |  |  |  |
| description[1] |  | Very high | High | Very high |  |  |  |
| Extraversion | 123.61 (18.07) | 65.74 (16.65) | 71.24 (15.63) | 100.40 (14.42) | 250.93 | < .001 | HC>3>(1 = 2) |
| d vs. norms[1] |  | -2.23 | -1.98 | -0.40 |  |  |  |
| description[1] |  | Very low | Very low | Slightly low |  |  |  |
| Openness | 121.27 (22.42) | 94.38 (17.04) | 99.85 (20.01) | 124.45 (16.57) | 46.56 | < .001 | (HC = 3)>(1 = 2) |
| d vs. norms[1] |  | -0.56 | -0.26 | 1.01 |  |  |  |
| description[1] |  | Moderately low | Slightly low | High |  |  |  |
| Agreeableness | 131.79 (18.24) | 131.59 (17.67) | 132.34 (17.25) | 130.28 (19.79) | .17 | .914 | HC = 1 = 2 = 3 |
| d vs. norms[1] |  | 0.07 | 0.12 | -0.001 |  |  |  |
| description[1] |  | Average | Average | Average |  |  |  |
| Conscientiousness | 126.33 (20.67) | 96.75 (17.33) | 126.48 (14.23) | 107.74 (19.53) | 49.75 | < .001 | (HC = 2)>3>1 |
| d vs. norms[1] |  | -1.36 | 0.30 | -0.71 |  |  |  |
| description[1] |  | Very low | Slightly high | Moderately low |  |  |  |

NEO-PI-R = Revised NEO Personality Inventory; d = between-group effect size according to Cohen's d

[1]SAD in comparison to Swedish norm data [67].

category ($\chi^2$ = 25.97,df = 4,p<.001,n = 211). SAD was deemed to be severe in 59% of the individuals in the *Prototypical* cluster as compared to 23% and 25% of the *Introvert-Conscientious* and *Instable-Open* clusters respectively. Also, as assessed in a subset of the sample, generalized

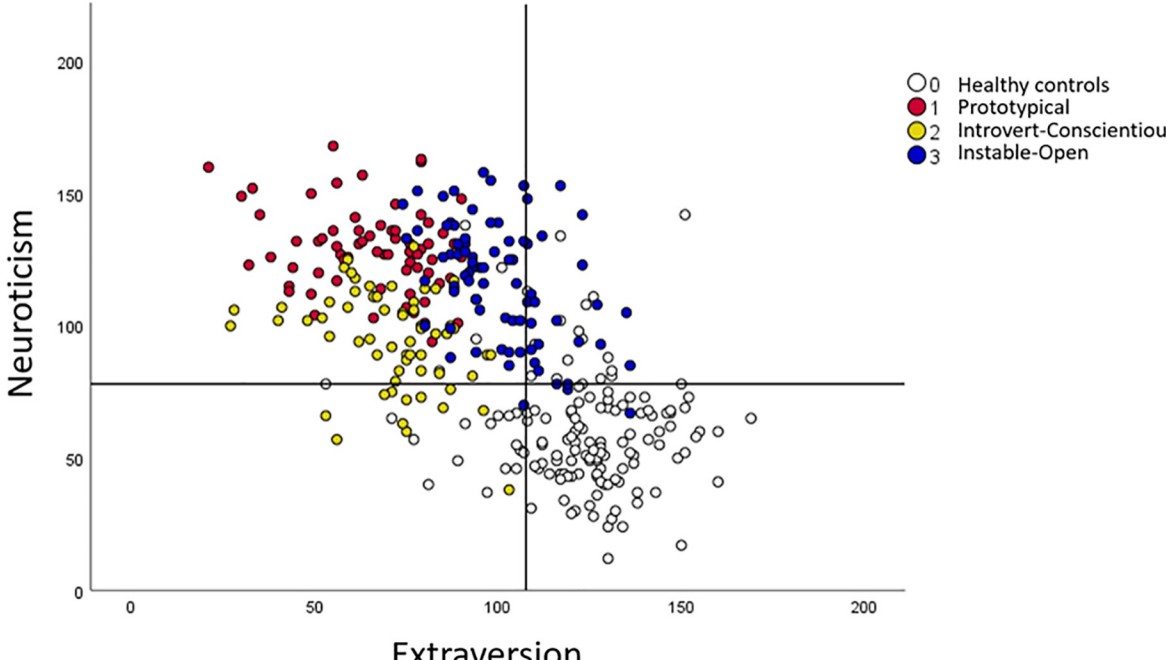

**Fig 2. Distribution of three clusters of social anxiety disorder patients and healthy controls along the neuroticism and extraversion dimensions.** The crosshair denotes Swedish norm values for neuroticism (M = 78.0, SD = 22.5) and extraversion (M = 107.6, SD = 20.7) respectively.

**Table 5. Mean values (SD) and ANOVA results on the six profiling variables in the three clusters of social anxiety disorder.**

| | 1 (n = 69) Prototypical | 2 (n = 62) Introvert-Conscientious | 3 (n = 79)[b] Instable-Open | F (2,210) | P | Post-hoc |
|---|---|---|---|---|---|---|
| Age | 30.86 (8.75) | 36.08 (12.05) | 31.73 (10.36) | 4.70 | .010 | (1 = 3)<2 |
| LSAS | 82.68 (19.73) | 74.45 (23.21) | 64.03 (20.97) | 14.31 | < .001 | 3<(1 = 2) |
| SIAS | 57.42 (11.54) | 52.85 (14.25) | 45.70 (13.95) | 14.65 | < .001 | 3<(1 = 2) |
| STAI-T[a] | 55.74 (6.50) | 46.58 (10.70) | 46.93 (12.43) | 11.34§ | < .001 | (2 = 3)<1 |
| Fear Speech | 75.94 (20.73) | 62.35 (28.81) | 63.10 (23.36) | 7.26[c] | .001 | (2 = 3)<1[c] |
| Distress Speech | 81.64 (19.17) | 76.44 (22.53) | 72.97 (20.99) | 2.97[c] | .053 | |

LSAS = Liebowitz Social Anxiety Scale; SIAS = Social Interaction Anxiety Scale

[a]STAI-T = State Trait Anxiety Inventory–Trait (data available for n = 136

§df = 2, 133)

[b]missing data for n = 1

[c]ANCOVA (df = 2,209) and planned simple contrasts.

SAD ($\chi^2$ = 8.70, df = 1, p = .003, n = 72) and avoidant personality disorder ($\chi^2$ = 19.42, df = 1, p<.001, n = 73) were more common in the *Prototypical* cluster than in the remainder of SAD patients. The percentages of generalized SAD/avoidant personality disorder in the three clusters were: 88/83% for *Prototypical*, 69/38% for *Introvert-Conscientious*, and 46/25% for *Instable-Open* SAD.

## Discussion

The current study compared personality traits, assessed with the NEO-PI-R and KSP instruments, between patients diagnosed with SAD and healthy controls and between different subtypes of SAD identified through cluster analysis. Overall, marked case-control differences in personality traits were noted on the NEO-PI-R Big Five personality dimensions, with the exception of agreeableness, and differences were also noted on the majority of facets and most KSP variables. Logistic regression analysis of NEO-PI-R showed that only neuroticism and extraversion remained significant independent predictors of SAD/control group when controlling for the effects of other predictors in the model. Two-step cluster analysis of the NEO-PI-R data yielded three clusters labelled *Prototypical* (33%), *Introvert-Conscientious* (29%), and *Instable-Open* (38%) based on their most noticeable features. *Prototypical* SAD had the most maladaptive personality profile and represented the most severe form of SAD as shown in further analyses.

Thus, the group comparisons indicated associations between SAD and several personality domains, but neuroticism and extraversion had the highest ability to discriminate between SAD patients and healthy controls. Only these two personality dimensions remained robust significant predictors of group (SAD/control) in the logistic regression analysis controlling for other predictors in the statistical model. Moreover, only neuroticism and extraversion yielded large between-group effect sizes when SAD patients were compared with Swedish normative data while a moderate effect was noted also for conscientiousness, being lower in patients. The current findings converge with previous studies reporting high neuroticism and low extraversion [16,19] as well as high KSP anxiety predisposition, detachment, and low socialization and social desirability [20] in patients with SAD. Similarly, studies using the TCI have noticed differences between SAD patients and controls with regard to harm avoidance and novelty seeking, frequently described as being related to neuroticism and/or extraversion [22–24]. Previous research also suggests that conscientiousness, agreeableness and openness show only weak associations with SAD when neuroticism and extraversion have been accounted for [71]. While elevated neuroticism has been demonstrated to be a common feature of many emotional disorders, low extraversion may be more specific for SAD [19,72].

Because the five broad dimensions are considered to be less powerful and less specific in the prediction or explanation of behavior as compared to facets [72,73], we also analyzed the lower-order traits. Both in comparison to healthy controls and normative data, we observed effect sizes of large magnitude for the majority of neuroticism and extraversion facets, including high self-consciousness-N4 and low assertiveness-E3, previously suggested to be specific features of SAD [72,74]. In the SAD group, low scores were noted on the positive emotion-E6 facet which may be a shared feature of SAD and major depression [72,75]. The SAD sample did not differ from norms with regard to excitement seeking-E5, mainly explained by high scores in the *Instable-Open* cluster. Congruently, previous studies have reported weak correlations between social anxiety and fun-seeking [74] and higher levels of excitement-seeking in SAD as compared to panic and post-traumatic stress disorder [72]. Other studies have also found associations between social anxiety and low trust-A1, competence-C1 and achievement-striving-C4 [17,76]. In the current study, SAD was associated with low competence-C1, self-discipline-C5, and openness to actions-O4 which may reflect neophobic behavior. Mixed effects on agreeableness facets, i.e. lower trust-A1 and altruism-A3, but higher straightforwardness-A2 and modesty-A5, were observed in the SAD patients compared with healthy controls, impeding significant group differences in the higher-order trait.

An additional goal was to elucidate subtypes of SAD derived from the Big Five personality dimensions. Two-step cluster analysis revealed three distinct personality types. Patients in the *Prototypical* cluster had significantly higher levels of neuroticism, and lower levels of conscientiousness than the other clusters. They also exhibited the lowest levels of extraversion and openness although differences on these variables were significant only in relation to the *Instable-Open* cluster. On NEO-PI-R facets, *Prototypical* patients manifested low openness to ideas-O5, as well as low trust-A1, competence-C1, achievement-striving-C4 and self-discipline-C5, i.e., traits associated with less adaptive pro-social attitudes and higher anxiety [76]. Profile analyses indicated that patients in this cluster had the highest levels of social anxiety symptom severity (LSAS) and significantly higher trait anxiety and fear during public speaking than both other clusters. On KSP variables they deviated on psychic anxiety, muscular tension, psychasthenia, and guilt. Thus, these patients can be described as the most severe subgroup with an anxious-introvert personality profile fitting the "prototypical" description of SAD that also has been identified in other cluster analytic studies, e.g. [39]. They could also be described as having a highly overcontrolled personality type [40]. However, the *Prototypical* cluster contained only about one third of the clinical sample, suggesting that considerable phenotypic variability is embedded in the SAD diagnostic category.

Individuals in the *Introvert-Conscientious* cluster, constituting 29% of the SAD sample, were characterized by significantly higher levels of conscientiousness (indistinguishable from healthy controls) and lower levels of neuroticism compared with the other clusters. Conscientiousness reflects a reasonable efficient need for achievement and self-discipline and individuals scoring low on this dimension may use poor coping strategies. Conversely, it could be argued that high conscientiousness represents a protective factor, possibly enhancing emotional stability. Notably, these individuals were still very introverted and scored low on openness (indistinguishable from the *Prototypical* cluster). Also, this cluster resembled the low impulsive type identified by Mörtberg and colleagues [39], considering their low levels of impulsiveness-N5 and very low levels of KSP-impulsivity. When compared to the other two clusters, *Introvert-Conscientious* patients manifested lower somatic anxiety, lower irritability, and comparable levels of guilt with the controls, as measured by the KSP scales. However, their levels of social anxiety, trait anxiety, public speaking fear and distress were still high although generally not as high as in the *Prototypical* cluster.

The *Instable-Open* cluster was the largest, representing 38% of the entire SAD sample. These patients had, by normative standards, very high levels of neuroticism but on openness

they were indistinguishable from the healthy controls and they had considerably higher openness values both in comparison to norm data and the other SAD clusters. This was particularly noticeable on the fantasy-O1 and feelings-O3 facets. They also emerged as a stand-out group with regard to extraversion. In a way, these individuals could be described as "anxious extraverts" although their level of extraversion was not quite on par with the healthy controls. Exceptions were noted for the E4-acitivity and E5-excitement seeking facets where *Instable-Open* patients and controls were indistinguishable, and this was also true for the impulsivity and monotony avoidance scales of the KSP. Relative to the other clusters, *Instable-Open* patients were characterized by lower detachment and higher impulsivity and monotony avoidance, i.e. KSP scales that are correlated with extraversion. Studies of temperament characteristics in SAD have similarly noted that a considerable portion, about 20–40% of patients, score comparatively high on novelty seeking, held to be one aspect of extraversion [39]. There are several reports of an atypical SAD subgroup with high novelty seeking and harm avoidance along with more impulsive decision making and risk-prone behavior like substance misuse, self-harm, aggression or unsafe sex practises [36–39]. Risk behaviors of this kind were not systematically assessed in the present study, making comparisons difficult, but it is noteworthy that patients in the *Instable-Open* cluster had even higher values on excitement-seeking-E5 in comparison to norms (M 18.2 vs. 14.4) but they did not differ from normative data on KSP-impulsivity. They also had significantly higher levels of self-discipline-C5 and lower levels of social anxiety and interaction anxiety than *Prototypical* SAD. Taken together, this appears incongruent with previous findings on the atypical anxious-impulsive SAD subtype [36–39], although it is possible that a subset of patients in the *Instable-Open* cluster had this profile.

To our knowledge, the Big Five personality dimensions have not previously been used to delineate empirically derived SAD subtypes. It remains to be tested if the present personality clusters differ qualitatively with respect to type of social fear as identified in factor analytic studies [35], or if they differ predominantly on quantitative measures. The present subtype data are partly consistent with the dimensional "continuum of severity" view, in that the *Prototypical* and *Instable-Open* cluster differed quantitatively on measures of social anxiety symptom severity. Also, the *Instable-Open* and the *Introvert-Conscientious* clusters could be differentiated on SAD severity measured with LSAS but not with regard to trait anxiety or public speaking fear and, between the two, levels of neuroticism were significantly higher in the *Instable-Open* cluster. These two clusters also had equal numbers of severe patients according to the clinical interviews. Thus, while *Prototypical* SAD stood out as the most severe cluster, the other two presented a more mixed pattern, not fitting clearly with a dimensional model. The current results suggested high overlap between *Prototypical* SAD and avoidant personality disorder that frequently has been described as a severe form of SAD [77]. Also, as suggested by the present data, the *Prototypical* cluster is probably most similar to the "generalized SAD" typology. Consistently, Stemberger and colleagues noted, in a smaller clinical sample, higher levels of neuroticism and lower levels of extraversion in patients with generalized as compared to specific social phobia [78].

It was evident that all three SAD clusters had higher levels of neuroticism and lower levels of extraversion in comparison with healthy controls as well as norm data, whereas the overlap was larger on the other personality variables. Extraversion and neuroticism also had the highest predictor importance in the cluster analysis. Interestingly, genetic and twin studies have suggested that social anxiety has a genetic basis that may be shared with extraversion and possibly also neuroticism [10,79]. The concept of shyness was initially rooted in the interaction between neuroticism and extraversion, i.e. individuals low on extraversion and high on neuroticism were characterized as being socially shy [80,81]. But individuals may also be highly introverted without showing excessive anxiety, i.e. shyness and introversion should not be

viewed as identical constructs. Several individuals in the *Introvert-Conscientious* cluster appeared very introverted together with relatively moderate levels of neuroticism and, conversely, several patients in the *Instable-Open* cluster were very anxious without being particularly introverted. Thus, many individuals in these two clusters do not exhibit a clear shyness profile. Patients in the *Prototypical* cluster could, however, be described as very shy, and perhaps these individuals exhibit the more severe and persistent form of temperamental shyness and social withdrawal that emerges during early infancy [82]. A strong neurobiological origin, including amygdala hyper-responsiveness, has been suggested for inhibited temperament of this kind [83].

## Limitations

Our findings should be interpreted with some limitations in mind. First, both SAD patients and controls were composed of Swedish participants in neuroimaging trials recruited through advertisements, which may have introduced selection biases and generalizability issues. One concern may be that relatively mildly affected SAD individuals were enrolled because patients with ongoing treatment were excluded, and individuals volunteering for research trials may differ from those being within the mental health care system, e.g., in terms of symptom severity, comorbidity, global functioning, and willingness to participate in research involving a public speaking challenge. However, the present sample had similar levels of social anxiety symptom severity, as measured with LSAS, as typically reported in clinical trials [84]. It should be noted that SAD cases with circumscribed performance fears were largely lacking in the present sample although they are not uncommon in the general population [85]. In comparison to the Swedish normative population for NEO-PI-R [67,86], the healthy control group had somewhat deviant values, suggesting imperfect representation of the general population, e.g. because of lower mean scores of neuroticism (70.4 vs. 78.0) and higher mean sores of extraversion (116.8 vs. 107.6) and openness (121.3 vs. 105.2). Thus, they could be described as having a "role model" rather than the more common "average" personality type reported by Gerlach and colleagues [44]. This may be expected since the control subjects volunteered for a research project and had to be free of SAD and other psychiatric disorders in order to be enrolled.

There are many viable alternatives, or complementary statistical methods, to the two-step cluster analysis used in the present trial. For example, regularized partial correlation networks [87] may be a fruitful approach to examine the network structure in personality data in future research. Moreover, the present data were collected in a neuroimaging research context lacking certain psychometric evaluations like inter-rater reliability of the clinical interviews. Diagnostic information on generalized SAD and avoidant personality disorder were available only for a subset of the sample and should therefore be interpreted with caution. Also, even though the NEO-PI-R and KSP instruments were filled out in the comfort of the participant's home, personality ratings could perhaps be biased by general distress levels or state effects in treatment-seeking individuals. Because personality assessments were only conducted at one time point, before neuroimaging and treatment, it is not known if the deviant personality traits predate SAD onset, influencing the expression of the disorder, or if the personality ratings are a consequence of the disorder. Longitudinal designs are needed to address this.

There is a need of further studies examining if the current personality differences are specific for SAD and if they are generalizable across epidemiological-clinical samples and culturally diverse populations. Also, the current SAD clusters should ideally be compared, not only on social anxiety symptom severity, but also on personality functioning, involving self (identity, direction) and interpersonal (empathy, intimacy) dimensions demonstrated to be impaired in anxiety disorders [88]. Assessment of personality functioning has been added to

the alternative diagnostic model for personality disorders in DSM-5. Depression levels were not included in the current analyses because different depression inventories were used across the trials. However, previous research has indicated that the relationship between social anxiety and depression is accounted for by approach-avoidance temperamental vulnerabilities [89]. Finally, future studies should examine how SAD personality heterogeneity is related to other clinical and biological factors like genetics [10], aversive learning experiences [78], cognitive biases [90], attachment styles [91], neuroimaging markers [92] and therapy outcome [18]. Interestingly, Mörtberg et al. noted that only 20% of patients in the prototypical inhibited cluster responded to CBT [39]. On the other hand, Stein and colleagues reported that escitalopram was equally effective in patients with more and less severe social anxiety symptoms and that the SSRI was effective across different SAD symptom dimensions [93]. In a long-term treatment outcome perspective, it is not known if personality variables are related to remission or relapse rates.

## Conclusions

While SAD, on a group level, is characterized by largely deviant scores on neuroticism and extraversion and their lower-order facets, the present results also point to considerable personality heterogeneity within the disorder. Only one third of the SAD patients fit well with the "anxious-introvert" (shy) personality profile typically associated with the condition. Indeed, SAD appears to be multidimensional and could be conceptualized as a spectrum disorder [94]. This may have important clinical and theoretical implications. For example, SAD personality subtypes may have different etiologies and it seems plausible that individuals exhibiting vastly different personality characteristics require different treatment strategies. Current CBT interventions, predominantly targeting neuroticism and behavioral avoidance, could be extended to better address maladaptive extraversion components like low levels of positive emotions, especially in the *Prototypical* and *Introvert-Conscientious* clusters. For example, such interventions may include behavioral activation, developed to treat anhedonia and low energy levels in depressed patients [95], or CBT augmented by a relational/social approach focus [96]. Personality assessment could improve clinical phenotyping and diagnostic precision, providing better understanding of the hierarchical structure of social anxiety in relation to other internalizing disorders or other conceptualizations like avoidant personality disorder [97]. Personality assessment could also enable recruitment of more homogenous samples e.g., in neuroimaging, genetic and treatment trials where sample sizes often are small. Finally, personality assessment could assist in treatment planning and response prediction, for example by informing on individual strengths and vulnerabilities that bear impact on the choice of psychotherapeutic techniques, pharmacological agents or their combination.

## Supporting information

**S1 Table. Comparisons of social anxiety disorder (SAD) patients and Healthy Controls (HC) on Revised NEO Personality Inventory facets.**
(DOCX)

**S2 Table. Correlations between Karolinska Scales of Personality items and the Revised NEO Personality Inventory dimensions in the social anxiety disorder group.**
(DOCX)

**S3 Table. Mean values (SD) and ANOVA results on the Revised NEO Personality Inventory facets in three clusters of social anxiety disorder (SAD) patients in comparison to**

**healthy controls (HC).**
(DOCX)

**S4 Table. Mean values (SD) and ANOVA results on the Karolinska Scales of Personality variables in the three clusters of social anxiety disorder (SAD) patients in comparison to healthy controls (HC).**
(DOCX)

## Author Contributions

**Conceptualization:** Mădălina Elena Costache, Tomas Furmark.

**Data curation:** Mădălina Elena Costache.

**Formal analysis:** Mădălina Elena Costache, Tomas Furmark.

**Funding acquisition:** Mats Fredrikson, Tomas Furmark.

**Investigation:** Mădălina Elena Costache, Andreas Frick, Kristoffer Månsson, Jonas Engman, Vanda Faria, Olof Hjorth, Johanna M. Hoppe, Malin Gingnell, Örjan Frans, Johannes Björkstrand, Jörgen Rosén, Iman Alaie, Fredrik Åhs, Clas Linnman, Kurt Wahlstedt, Maria Tillfors, Ina Marteinsdottir, Mats Fredrikson, Tomas Furmark.

**Supervision:** Mats Fredrikson, Tomas Furmark.

**Writing – original draft:** Mădălina Elena Costache, Tomas Furmark.

**Writing – review & editing:** Mădălina Elena Costache, Andreas Frick, Kristoffer Månsson, Jonas Engman, Vanda Faria, Olof Hjorth, Johanna M. Hoppe, Malin Gingnell, Örjan Frans, Johannes Björkstrand, Jörgen Rosén, Iman Alaie, Fredrik Åhs, Clas Linnman, Kurt Wahlstedt, Maria Tillfors, Ina Marteinsdottir, Mats Fredrikson, Tomas Furmark.

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
