## [Decision Letter · Decision Letter 0]

15 Jan 2020

PONE-D-19-29839

Higher- and Lower-order Personality Traits and Cluster Subtypes in Social Anxiety Disorder

PLOS ONE

Dear Dr Furmark,

Thank you for submitting your manuscript to PLOS ONE. After careful consideration, we feel that it has merit but does not fully meet PLOS ONE’s publication criteria as it currently stands. Therefore, we invite you to submit a revised version of the manuscript that addresses the points raised during the review process.

We would appreciate receiving your revised manuscript by Feb 29 2020 11:59PM. To enhance the reproducibility of your results, we recommend that if applicable you deposit your laboratory protocols in protocols.io, where a protocol can be assigned its own identifier (DOI) such that it can be cited independently in the future. For instructions see: http://journals.plos.org/plosone/s/submission-guidelines#loc-laboratory-protocols

We look forward to receiving your revised manuscript.

Kind regards,

Frantisek Sudzina

Academic Editor

PLOS ONE

2. Thank you for your ethics statement in your manuscript:

"All studies were ethically approved and all participants provided written informed consent."

Reviewers' comments:

Reviewer's Responses to Questions

**Comments to the Author**

1. Is the manuscript technically sound, and do the data support the conclusions?

Reviewer #1: Yes

Reviewer #2: Partly

2. Has the statistical analysis been performed appropriately and rigorously? 

Reviewer #1: Yes

Reviewer #2: Yes

3. Have the authors made all data underlying the findings in their manuscript fully available?

Reviewer #1: Yes

Reviewer #2: Yes

4. Is the manuscript presented in an intelligible fashion and written in standard English?

Reviewer #1: Yes

Reviewer #2: Yes

5. Review Comments to the Author

Reviewer #1: This study compares personality traits measures (using NEO and KSP questionnaires) between SAD patients and controls, and explores the presence of latent empirical clusters based on the personality profiles reported among clinical subsample. It has been conducted in a large sample of patients diagnosed with SAD (n=265) and controls (n=164).

My overall assessment: this is a really well-written, interesting and novel study that is a delight to read. My minor comments are only for completeness of reporting.

Background: This section is well argued adding a great deal of new insight to this topic. It also logically generates the aims.

I only suggest including the empirical hypothesis of the study based on the theoretical background.

Methods:

a. It is not convenient to report mean and SD as “mean+-SD”. It is preferable to use alternative formats, such as “mean (SD)”.

b. Recruitment for SAD group was between 1998 and 2018. What was the recruitment time for the control sub-sample?

c. It is quite surprising none of the controls fulfilled the screening criteria for SAD or any other psychiatric condition. What were the measures used to carry out the screening?

d. Last paragraph in Page 8 should be included as a part of the results section. In addition, authors report the significant correlation estimates between the patients’ age with some NEO-PI-R or KSP scales. Due the strong association between significant test for correlation coefficients and sample size, p-values are low, but effect sizes are really poor. This should be outlined. In addition, I do not understand why the authors mean with weak-uncorrected correlations

e. For each of the scales used in the study, the internal consistency in the sample (Cronbach’s-alpha) must be included

f. Statistical analysis: I suggest using a procedure for correcting the increase in Type-I error due to the statistical multiple comparisons different to Bonferroni-method. Alternative procedures (such as Finner, Holmes or Simes) have evidenced to be preferable alternatives.

g. Statistical analysis: the formula for applying corrective methods for multiple comparisons is not required (use a cite-reference).

h. Statistical analysis: I suggest to change Bonferroni post-hoc comparisons in the ANOVA for other more powerful methods, since DNS.

Discussion: Well presented and logical. It does not overstate the findings.

Reviewer #2: Analysis conducted explain the main objetive of the paper. However, it could be appreciated that to approach relationships between variables, Regularized Partial Correlation Networks (RPCNs) could be selected as the analytic framework. As it had been shown in psychopathology and personality research, RPCNs are closely similar to Structural Equation Modeling but allow exploratory models to take place. They combine the undirected graphs (with individual entities as ‘nodes’ and relations between them as ‘edges’) with the multivariate statistics (i.e. correlation) frameworks (Epskamp & Fried, 2018) assuming nodes to be random variables and edges as unobserved and needed to be estimated. Thus, they allow estimating relationships between relatively large sets of variables in an exploratory manner.It could be interesting in stead or complementary to cluster analysis. RPCNs take partial correlation coefficients and apply regularization techniques to then display them as graphical networks between variables via the Glass algorithm (Epskamp, Borsboom, & Fried, 2018).

6. PLOS authors have the option to publish the peer review history of their article (what does this mean?). If published, this will include your full peer review and any attached files.

Reviewer #1: No

Reviewer #2: No

---

## [Author Response · Author response to Decision Letter 0]

27 Mar 2020

Reviewer #1: This study compares personality traits measures (using NEO and KSP questionnaires) between SAD patients and controls, and explores the presence of latent empirical clusters based on the personality profiles reported among clinical subsample. It has been conducted in a large sample of patients diagnosed with SAD (n=265) and controls (n=164).

My overall assessment: this is a really well-written, interesting and novel study that is a delight to read.

Response: Thank you. We really appreciate these encouraging comments.

My minor comments are only for completeness of reporting.

Background: This section is well argued adding a great deal of new insight to this topic. It also logically generates the aims. I only suggest including the empirical hypothesis of the study based on the theoretical background.

Response: We agree and we have now added our hypotheses to the last section of Introduction (p.7). 

Methods:

a. It is not convenient to report mean and SD as “mean+-SD”. It is preferable to use alternative formats, such as “mean (SD)”.

Response: We have now changed this to “mean (SD)” as suggested.

b. Recruitment for SAD group was between 1998 and 2018. What was the recruitment time for the control sub-sample?

Response: The recruitment time for the control sample was the same as for patients. The first controls were assessed (KSP variables only) in 1998 and later published in Tillfors et al. Am J Psychiatry, 2001. We have tried to explain this, albeit in an “economical” way, in Methods (p.7)

“All participants were volunteers in neuroimaging treatment trials, data being collected from 1998 to 2018, as described elsewhere [46-54].”

c. It is quite surprising none of the controls fulfilled the screening criteria for SAD or any other psychiatric condition. What were the measures used to carry out the screening?

Response: We agree that, in a few cases, it was a bit surprising that controls did not fulfil the criteria for a psychiatric condition – considering that they were “outliers” on personality scores and somewhat elevated social anxiety levels could be detected for some. However, the level of impairment was not substantial and none of these cases, or any other control, fulfilled the criteria for a psychiatric disorder. This was assessed with the same measures as for patients, i.e. the anxiety disorders section of Structured Clinical Interview for DSM-IV and the Mini International Neuropsychiatric Interview. There was one exception with regards to screening, in the first study (Tillfors et al. Am J Psychiatry, 2001) the full SCID-I and SCID-II interviews were conducted. This has now been clarified in Methods (p. 8).

d. Last paragraph in Page 8 should be included as a part of the results section. In addition, authors report the significant correlation estimates between the patients’ age with some NEO-PI-R or KSP scales. Due the strong association between significant test for correlation coefficients and sample size, p-values are low, but effect sizes are really poor. This should be outlined. In addition, I do not understand why the authors mean with weak-uncorrected correlations

Response: Thank you. We have now moved this to the Results section (first paragraph, p. 11). We have deleted the word “uncorrected” and emphasize that these correlations are indeed weak. We think we are saying the same thing as the reviewer here.

e. For each of the scales used in the study, the internal consistency in the sample (Cronbach’s-alpha) must be included

Response: As the personality instruments are the prime focus of this paper, we have added internal consistency values for the NEO-PI-R and KSP (Methods, p. 9). Unfortunately, data on profiling variables (LSAS, SIAS, STAI-T, fear and distress ratings) were not digitally stored in a uniform way over the 20-year data collection period, i.e. data were stored mainly as summed scores. Therefore, we could not determine alpha values. However, the scales are established, validated and widely used. In the discussion, we apologize (limitations p. 29), that certain psychometric data are unfortunately lacking.

f. Statistical analysis: I suggest using a procedure for correcting the increase in Type-I error due to the statistical multiple comparisons different to Bonferroni-method. Alternative procedures (such as Finner, Holmes or Simes) have evidenced to be preferable alternatives.

Response: We agree that the Bonferroni-method is too conservative when the number of comparisons increases, and in the present study this is especially the case when comparing patients and controls on NEO-PI-R facets (30 comparisons) and KSP variables (15 comparisons). Thus, in the revised manuscript we applied Holm adjusted alpha levels, as suggested by the reviewer for facets and KSP items. This yielded only one more significant result – verbal aggression now also deviated significantly between SAD and healthy controls. Consequently, wording has been changed in Methods (statistical analyses p. 10), Results (p. 14) and in the notes to Table 3 and Supplementary S1 table. For the full Big Five dimensions only five comparisons were made and the significant effects were robust so here we kept the Bonferroni method.

g. Statistical analysis: the formula for applying corrective methods for multiple comparisons is not required (use a cite-reference).

Response: We have removed the formula as suggested. We have also added (Methods p.10) that we used Holm adjusted alpha levels – see above.

h. Statistical analysis: I suggest to change Bonferroni post-hoc comparisons in the ANOVA for other more powerful methods, since DNS.

Response: Since the pattern or results was very clear, and none of the conclusions would be altered with another post-hoc test, we would prefer keeping the Bonferroni post-hoc comparisons. While we acknowledge that there are other (and presumably more sensitive) post hoc alternatives, the Bonferroni method is still valid and a widely used post hoc test. 

Discussion: Well presented and logical. It does not overstate the findings.

Response: Thanks, we are very grateful for the thoughtful comments and suggestions for improvement. We added a new reference to the final paragraph (p. 31) highlighting current cluster analytic research on differences and similarities between social anxiety disorder and avoidant personality disorder:

Frandsen FW, Simonsen S, Poulsen S, Sørensen P, Lau ME. Social anxiety disorder and avoidant personality disorder from an interpersonal perspective. Psychol Psychother Theory, Res Pract 2020;93:88–104.

Reviewer #2: Analysis conducted explain the main objetive of the paper. However, it could be appreciated that to approach relationships between variables, Regularized Partial Correlation Networks (RPCNs) could be selected as the analytic framework. As it had been shown in psychopathology and personality research, RPCNs are closely similar to Structural Equation Modeling but allow exploratory models to take place. They combine the undirected graphs (with individual entities as ‘nodes’ and relations between them as ‘edges’) with the multivariate statistics (i.e. correlation) frameworks (Epskamp & Fried, 2018) assuming nodes to be random variables and edges as unobserved and needed to be estimated. Thus, they allow estimating relationships between relatively large sets of variables in an exploratory manner.It could be interesting instead or complementary to cluster analysis. RPCNs take partial correlation coefficients and apply regularization techniques to then display them as graphical networks between variables via the Glass algorithm (Epskamp, Borsboom, & Fried, 2018).

Response: Thank you. The reviewer raises an important topic. There are indeed several possible analytic procedures to consider, e.g. with regards to classification, and there is an ongoing debate whether network analyses, viewing symptoms as dynamic systems of mutually interacting variables, are more fruitful than a latent variable approach. In the present study, we chose a rather “simple” method, i.e. a two-step cluster analysis, which is incorporated in statistical packages like SPSS. This is, however, an established, validated and widely used method, and there are indeed several previous cluster analytic studies of SAD that facilitate between-study comparisons. However, we agree with the reviewer that the Regularized Partial Correlation Network (RPCN) is one of many interesting methods, and there have been a few studies using this in SAD. For example, Rodebaugh and colleagues (Thomas L. Rodebaugh et al. Does Centrality in a Cross-Sectional Network Suggest Intervention Targets for Social Anxiety Disorder? J Consult Clin Psychol 2018; 86: 831–8442018) used RPCN to evaluate whether there are central interconnected symptoms of social anxiety that also are the most important treatment targets. Interpretations were, however, not straightforward. Apparently there are several caveats, and different central indices to consider. Because we feel that this is a topic for future research and development, and because we don’t master this method in the first place, we would prefer not to change our current analytic approach. We have, however, added the following to the discussion (limitation) section together with one reference and we hope that the reviewer is satisfied with this addition:

“There are many viable alternatives, or complementary statistical methods, to the two-step cluster analysis used in the present trial. For example, regularized partial correlation networks [87] may be a fruitful approach to examine the network structure in personality data in future research.”

Added reference: 

Epskamp S, Fried EI. A tutorial on regularized partial correlation networks. Psychol Methods 2018; 23: 617–634.

---

## [Decision Letter · Decision Letter 1]

9 Apr 2020

Higher- and Lower-order Personality Traits and Cluster Subtypes in Social Anxiety Disorder

PONE-D-19-29839R1

Dear Dr. Furmark,

We are pleased to inform you that your manuscript has been judged scientifically suitable for publication and will be formally accepted for publication once it complies with all outstanding technical requirements.

With kind regards,

Frantisek Sudzina

Academic Editor

PLOS ONE

Reviewers' comments:

Reviewer's Responses to Questions

**Comments to the Author**

1. If the authors have adequately addressed your comments raised in a previous round of review and you feel that this manuscript is now acceptable for publication, you may indicate that here to bypass the “Comments to the Author” section, enter your conflict of interest statement in the “Confidential to Editor” section, and submit your "Accept" recommendation.

Reviewer #1: All comments have been addressed

2. Is the manuscript technically sound, and do the data support the conclusions?

Reviewer #1: Yes

3. Has the statistical analysis been performed appropriately and rigorously? 

Reviewer #1: Yes

4. Have the authors made all data underlying the findings in their manuscript fully available?

Reviewer #1: Yes

5. Is the manuscript presented in an intelligible fashion and written in standard English?

Reviewer #1: Yes

6. Review Comments to the Author

Reviewer #1: The authors have been responsive to the reviewers' comments. All comments have been addressed thoroughly which further increased the manuscript's quality and comprehensibility. I have not additional suggestions and I would recommend the paper for publication. Congratulations on a comprehensive, interesting and very well designed paper

7. PLOS authors have the option to publish the peer review history of their article (what does this mean?). If published, this will include your full peer review and any attached files.

Reviewer #1: No

---

## [Editor Report · Acceptance letter]

16 Apr 2020

PONE-D-19-29839R1 

Higher- and Lower-order Personality Traits and Cluster Subtypes in Social Anxiety Disorder 

Dear Dr. Furmark:

I am pleased to inform you that your manuscript has been deemed suitable for publication in PLOS ONE. Congratulations! Your manuscript is now with our production department. 

With kind regards,

on behalf of

Dr. Frantisek Sudzina 

Academic Editor

PLOS ONE